

# Conspicuous colours reduce predation rates in fossorial uropeltid snakes

Vivek Philip Cyriac and Ullasa Kodandaramaiah

IISER-TVM Centre for Research and Education in Ecology and Evolution and School of Biology,
Indian Institute of Science Education and Research Thiruvananthapuram,
Thiruvananthapuram, Kerala, India

## ABSTRACT

Uropeltid snakes (Family Uropeltidae) are non-venomous, fossorial snakes that are found above ground occasionally, during which time they are exposed to predation. Many species are brightly coloured, mostly on the ventral surface, but these colours are expected to have no function below the ground. Observations have shown that the cephalic resemblance (resemblance to heads) of uropeltid tails may direct attacks of predators towards the hardened tails, thereby potentially increasing handling times for predators. Experiments have also shown that predators learn to avoid prey that are non-toxic and palatable but are difficult to capture, hard to process or require long handling time when such prey advertise their unprofitability through conspicuous colours. We here postulate that uropeltid snakes use their bright colours to signal long handling times associated with attack deflection to the tails, thereby securing reduced predation from predators that can learn to associate colour with handling time. Captive chicken experiments with dough models mimicking uropeltids indicate that attacks were more common on the tail than on the head. Field experiments with uropeltid clay models show that the conspicuous colours of these snakes decrease predation rates compared to cryptic models, but a novel conspicuous colour did not confer such a benefit. Overall, our experiments provide support for our hypothesis that the conspicuous colours of these snakes reduce predation, possibly because these colours advertise unprofitability due to long handling times.

# INTRODUCTION

Conspicuous colourations (colours that contrast with the background) are prevalent in the animal kingdom and have fascinated biologists for a very long time, prompting them to seek a functional explanation for such colours (*Poulton, 1890*; *Cott, 1940*; *Edmunds, 1974*; *Cuthill et al., 2017*). Such colours may be part of mate-choice or other intra-specific signals, or may be involved in predator avoidance, e.g. aposematism and mimicry (*Caro & Allen, 2017*). Over the last few decades, these theories have been widely tested, leading to a good understanding of bright conspicuous colours in animals (*Gamberale-Stille & Tullberg, 1999*; *Speed, 2001*; *Pfennig, Harcombe & Pfennig, 2001*; *Tullberg, Merilaita & Wiklund, 2005*; *Stevens, Stubbins & Hardman, 2008*).

Corresponding author
Vivek Philip Cyriac,
vivek.philip14@iisertvm.ac.in

However, the presence of conspicuous colours in many animals still remains unexplained (e.g. contrasting colours in mammals (*Caro, 2009*), colourations in many frogs (*Bordignon et al., 2018*) and caecilians (*Wollenberg & Measey, 2009*)) with research on new systems revealing novel insights onto the role of such colourations (*Rößler et al., 2019*). Subterranean organisms which live in conditions devoid of light tend to have reduced vision and lose pigmentations (*Culver & Pipan, 2009*). Yet, several species of fossorial snakes that spend most of their time underground exhibit bright colourations (e.g. members of the genus *Cylindrophis*, *Anomochilus*, *Atractus*. etc.) (*Greene, 1988*). Colours, especially those produced via pigments, are generally used as visual stimuli for intra-specific or inter-specific interactions and thought to be costly to produce, therefore, the presence of bright colours in subterranean animals is thought to be adaptive (*Wollenberg & Measey, 2009*). However, the role of conspicuous colouration in fossorial reptiles remains unexplored.

Uropeltid snakes are a family of fossorial snakes comprising ca. 55 species from South Asia (*Cyriac & Kodandaramaiah, 2017*). Most uropeltid species are ornamented on the ventral surface with bright conspicuous colouration, with the majority possessing varying degrees of yellow while a few species (ca. 5) possess red (Fig. 1). Although these snakes spend most of their time underground, they occasionally come to the surface during the monsoons (*Rajendran, 1985*). Uropeltids are mostly nocturnal but can be found actively moving close to the surface during early mornings and late evenings (*Rajendran, 1985*) during which times they are exposed to above ground predators (*Rajendran, 1985*; *Gans, 1986*; *Kumara & Chaitra, 2001*). Uropeltid snakes have characteristic morphologies adapted for burrowing into soil: a narrow head and a short distinctive tail which is tapering or rounded (*Smith, 1943*). The tails of many species appear obliquely cut, with hard carinate scales (Fig. S1). When attacked by predators, these snakes conceal their heads between their body coils and display their tails along with the conspicuously coloured ventral surface (*Gans, 1986*) (Fig. 2). The tail in uropeltid snakes, being short and rounded, resembles the head. This cephalic resemblance was found to deflect attacks of avian predators to the reinforced tail, thus allowing the snake to escape unharmed in most cases after multiple attacks oriented towards the tail (*Gans, 1986*). Such defenses would also potentially increase the handling time needed for predators to capture, kill and consume the prey.

Optimal foraging theory predicts that predators should avoid prey with increased handling times, especially when prey with lower handling times are abundant alternatives (*Charnov, 1976*; *Krebs et al., 1977*). Studies suggest that predators can learn to avoid unprofitable prey that are difficult to capture (*Hancox & Allen, 1991*; *Pinheiro, 1996*; *Pinheiro et al., 2016*), hard to process (*Wang et al., 2018*) or require long handling time (*Cyriac & Kodandaramaiah, 2019*) when such prey possess conspicuous colours, even if the prey are non-toxic and palatable (*Mappes, Marples & Endler, 2005*). We here postulate that bright colours in uropeltid snakes reduce predation, possibly by signalling long handling times associated with attack deflection to the tails. We first show that the yellow and red ventral colourations in uropeltid snakes are conspicuous to birds, the main predators of these snakes. In experiments involving captive chickens attacking dough

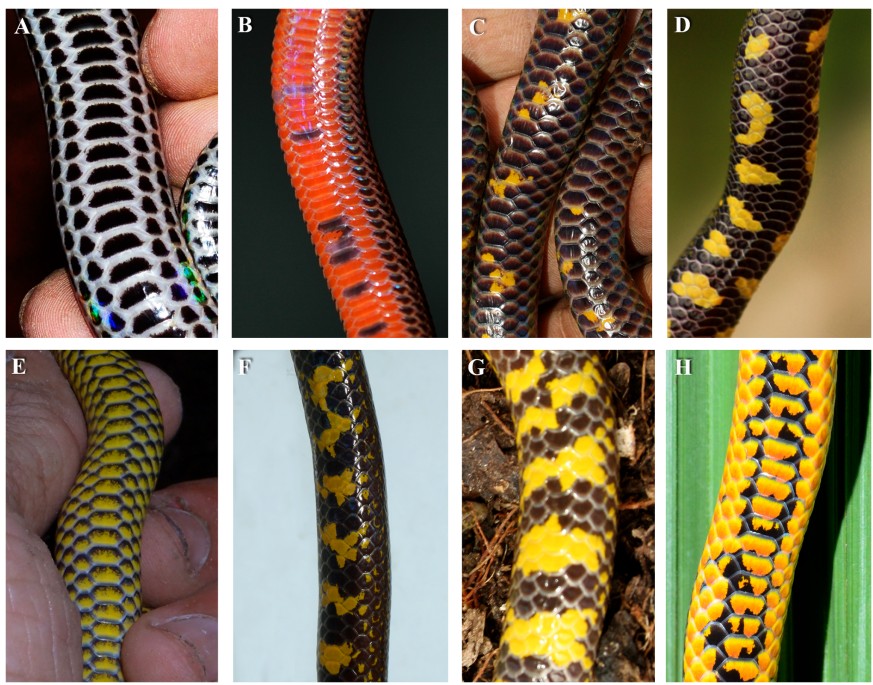

**Figure 1 Ventral colouration in different species of uropeltid snakes.** (A) *Melanophidium punctatum.* (B) *Teretrurus cf. sanguineus* from BBTC tea plantations. (C) *Plectrurus guentheri.* (D) *Uropeltis liura* from BBTC tea plantations. (E) *Uropeltis maculata.* (F) *Uropeltis* sp. (G) *Uropeltis* sp. from BBTC tea plantations. (H) *Uropeltis* sp. from BBTC tea plantations. Photo credit: (A), (G), (H) Umesh P.K. and (B)–(F) Vivek Philip Cyriac.

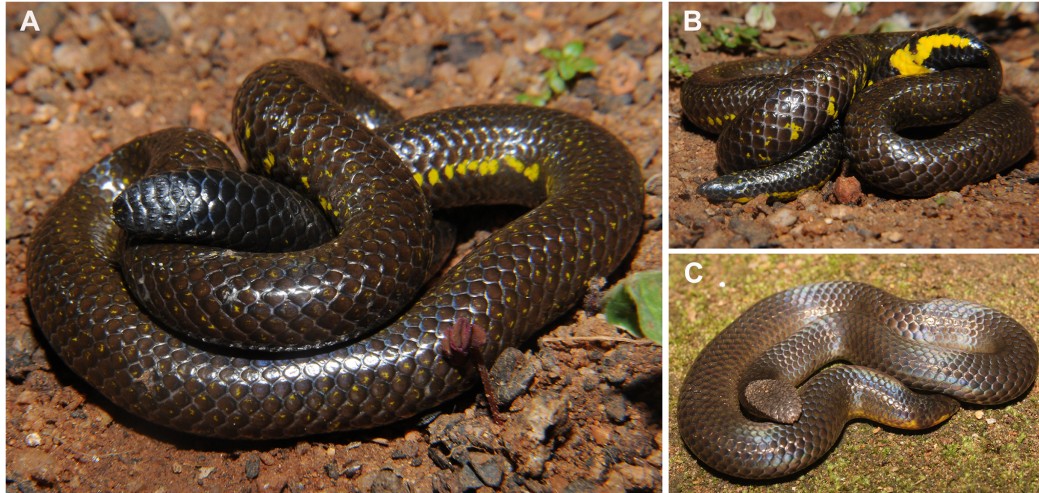

**Figure 2 Tail display in uropeltid snakes.** (A) *Uropeltis cf. ellioti* coiled with only the tail visible. (B) *U. cf. ellioti* displaying the tail which is laterally lined by a thick yellow band. (C) *U. cf. arcticeps* displaying the tail and hiding the head under its coils. Photo credit: Vivek Philip Cyriac.

models resembling uropeltid snakes, we show that attacks are directed more often towards the tail than the head. In field experiments with clay models that resemble uropeltid snakes, we show that the bright yellow and red colours of wild uropeltids decrease

predation by native birds. We conclude that the bright colours of these snakes reduce avian predation, and that these colours possibly play a role in advertising long handling time.

## MATERIALS AND METHODS

### Study system and area

We conducted our field experiment in the tea plantations of the Bombay Burma Trading Corporation (BBTC) (8.52°N, 77.40°E and 8.56°N, 77.25°E) in Tirunelveli district of Tamil Nadu, India. The BBTC tea plantations cover an area of ca. 34 km$^2$ with fragments of natural forest vegetation left as windbreaks between patches of plantations. The area harbours at least five species of uropeltid snakes (*Rajendran, 1985*) that vary in their colourations—*Melanophidiumn punctatum* which has a bluish iridescent colour, *Teretrurus cf. sanguineus* which is black dorsally and red ventrally, and three species of *Uropeltis* which are usually black dorsally with variable amounts of yellow colouration ventrally. Within this community, we focused on two species, *Uropeltis liura* and *Teretrurus cf. sanguineus*, that co-occur and are found in high abundance in the BBTC tea plantations.

### Avian visual modelling and conspicuousness against leaf litter background

We laid out four transects, each spanning 200 m in length, in four windbreak forest patches (Kakachi-1, Kakachi-2, Cullinia forest and Manjolai). Each transect was subdivided into 25 sub-transects perpendicular to the main transect at every eight meters, alternating in direction. From each sub-transect, we collected leaf litter samples representing the background in which the snakes are usually found. Leaf litter was obtained by randomly throwing a 30 × 30 cm cardboard frame thrice onto the forest floor, each time collecting 10–15 leaf samples. We measured the spectral reflectance measurements from five to six leaves representing the different visibly identifiable shades of all species from each sub-transect. We also measured the spectral reflectance of the ventral colourations of one individual each of *U. liura* and *Teretrurus cf. sanguineus* using a reflectance probe connected to a xenon light source (PX-2; Ocean Optics, Florida, USA) and a spectrophotometer (Maya 2000; Ocean Optics, Largo, FL, USA). We took three readings each on the dorsal and ventral surface of every leaf and ten readings from different locations on the ventral surface of both snake species. We calculated the mean reflectance of the leaf litter for the four transects and the two species of snakes after interpolating the raw reflectance values into one nanometer bins between 300 and 700 nanometer.

We modelled the ability of birds to distinguish the colouration of the two snake species against the leaf litter background using the receptor noise limited (RNL) model (*Vorobyev & Osorio, 1998*) in the R package *Pavo v. 2.1.0* (*Maia et al., 2019*). The RNL model assumes that colour vision is based on opponent interaction and models the ability to discriminate colours in relation to the noise in the opponent channel. The primary predators of uropeltid snakes are terrestrial birds such as junglefowl (*Gallus sonneratii*), spurfowl (*Galloperdix spadicea* and *Galloperdix lunulata*) and peafowl (*Pavo cristatus*) (*Rajendran, 1985*). Thus, we used the relative proportion of cone types of 1:2:4:4 for

domestic chickens (*Gallus gallus*) (*Kram, Mantey & Corbo, 2010*) and 0.92:1:0.81:0.54 for peafowl (*Hart, 2002*) representing the Violet-sensitive (VS) visual system of birds to model the colour contrasts. We also modelled the colour contrasts using the relative proportion of cone types of 1:0.99:0.71:0.37 for blue tits (*Cyanistes caeruleus*) (*Hart et al., 2000*), which represent the UV-sensitive (UVS) visual system. The relative proportion of cone types representing the short wavelength sensitive, medium wavelength sensitive, long wavelength sensitive and UVS cone type, respectively, were used with a Weber fraction of 0.06 (*Olsson, Lind & Kelber, 2015*) modelled under forest shade illumination. The results of the RNL model are summarised as $\Delta S$. When $\Delta S = 1$ the two stimuli are just noticeable, or 1 'just noticeable difference'. Following *Siddiqi et al. (2004)*, we considered a $\Delta S$ values above 3.00 to be easily distinguishable from the background, values between 1.00 and 3.00 to be indistinguishable except under optimal light conditions and values below 1.00 as a threshold for the colour to be indistinguishable from the background.

## Predation rates on snake models in field

To test if avian predators avoid ventral colourations of uropeltids, we prepared 500 snake models using non-toxic brown pre-coloured clay (Play clay; Uday Industries[TM], Goregaon, India) of five treatments varying in their colourations and resembling uropeltid snakes in size and shape. We placed these models in the four transects (Kakachi-1, Kakachi-2, Cullinia forest and Manjolai) and recorded predation rates. There were 100 models each of five treatments: (1) black dorsal with red ventral colouration representing *Teretrurus. cf. sanguineus*, (2) black dorsal with yellow ventral colouration representing *U. liura*, (3) black dorsal with orange ventral colouration representing a conspicuous novel colour not found in uropeltid snakes of the region (4) completely black models and (5) completely brown models representing many of the more cryptic snakes in the region. We used acrylic paints (Camel Fabrica[TM], Mumbai, India) for the five treatments. We replicated the yellow and red colours by mixing several combinations of paints, measuring their reflectance spectra and modelling them according to the RNL model using avian visual systems (Supplementary Material S1). We chose the paint combinations that produced the lowest $\Delta S$ values when compared with the spectral reflectance of the two snakes (see model design in Supplementary Methods for more details). The five treatments were placed in random order in each sub-transect at two meter intervals. Models were collected after 84–86 h and avian predation experienced by the clay replicas was scored based on predation marks on the clay models (Fig. S2). Avian predation was identified by the characteristic V-shaped, U-shaped or conical peck marks (*Willink et al., 2014*). We further confirmed these marks on the snake models by comparing them to images of confirmed avian attacks that we previously obtained by direct observation of jungle babblers (*Turdoides striata*) and indigenous domestic chickens attacking clay snake models. We conducted two predation trials, one in June 2017 and another in July 2017 with 500 models in each set (thus a total of 1,000 models) during the monsoons, when uropeltid snakes are known to be active above ground.

## Captive bird experiment

To test if the cephalic resemblance of the tail of uropeltid snakes deflects attacks of birds to the tail, we analysed data from another experiment that was designed to test whether birds could learn handling times (*Cyriac & Kodandaramaiah, 2019*). In this experiment, we used 26 chickens as predators, each uniquely identifiable, maintained in a small enclosure by a chicken farmer in Vithura (Kerala, India). The prey consisted of models that resemble uropeltid snakes (a tapering head and a rounded tail) made of wheat dough and brown food colour, presented on 'S' shaped yellow or brown coloured paper. Chickens were first trained to enter a 100 cm cubical arena and feed on small dough pieces scattered randomly in the experimental arena (acclimatization phase). Once they were acclimatised to the arena, in the next phase, we randomly divided the chickens into two groups and introduced a single dough model, fixed on yellow or brown paper, placed in the centre of the arena. Half of the dough models were baked, which increased the handling time of the chickens, while the remaining half were left unbaked. One group of chickens received the baked models on yellow paper and the unbaked models on brown paper, while the other group of chickens received both models on brown paper. Thus, we expected that chicken in the colour-associated group, where baked and unbaked models were provided on different colours, would learn the associated handling time and avoid baked models while the colour-unassociated group would show no preference. All chickens underwent 10 trials, during which they received a total of five baked and five unbaked models in random order (*Cyriac & Kodandaramaiah, 2019*). During this experiment, we also recorded the position of the first attacks by the chickens on the models as being on the head, mid-body or the tail.

## Ethical note

All applicable international, national and/or institutional guidelines for the care and use of animals were followed. The experimental protocol was approved by the Institutional Animal Ethics Committee of Indian Institute of Science Education and Research Thiruvananthapuram. The field experiments were done with the permission of Bombay Burma Trading Corporation, Limited, which owns the lands. This experiment does not require clearance from any Ethics committee.

## Analyses

All analyses were carried out in R 3.3.2 (*R Core Team, 2016*). To test for the effect of colour and random factors on the frequency of attacks on the snake models in the field experiment, we built generalized linear mixed models (GLMM) with the five treatments as fixed factors and transect, sub-transect, model sequence and batch (Table S2) as random factors using a binomial logit link function using the package *lme4 v. 1.1–12* (*Bates et al., 2015*). We compared the fit of this model with that of a null model using a likelihood ratio test. We followed the GLMM analysis by performing a post hoc test with Tukey contrasts to test for homogeneity across groups using the package *multcomp v. 1.4–8* (*Hothorn, Bretz & Westfall, 2008*). Further, we used the *G*-test using the package *RVAideMemoire v. 0.9–69* (*Hervé, 2014*) to check for differences in the position of first
**Table 1** Modelling the conspicuousness (ΔS) of ventral colouration in *Uropeltis liura* and *Teretrurus cf. sanguineus* against leaf background from the four transects in BBTC tea plantation for different avian predators.

| Transect | Avian predator | Kakachi-1 | Kakachi-2 | Cullinia | Manjolai | Mean ± SD |
|---|---|---|---|---|---|---|
| *Uropeltis liura* | Domestic chicken | 8.8676 | 9.1886 | 7.3817 | 6.8744 | 7.95 ± 1.25 |
| | Peafowl | 10.9248 | 11.6484 | 9.0312 | 8.6232 | 10.06 ± 1.46 |
| | Blue tit | 31.6323 | 37.6373 | 31.7853 | 28.9963 | 32.51 ± 3.65 |
| *Teretrurus cf. sanguineus* | Domestic chicken | 6.5345 | 7.4539 | 5.3492 | 5.5873 | 6.23 ± 0.96 |
| | Peafowl | 9.1048 | 10.3385 | 7.4867 | 7.7093 | 8.66 ± 1.33 |
| | Blue tit | 37.8320 | 44.1488 | 38.0672 | 35.5731 | 38.90 ± 3.67 |

Note:
ΔS values >3.00 are considered as easily distinguishable, values between 1.00 and 3.00 are distinguishable only under optimal light conditions and values <1.00 are indistinguishable from the background.

attacks by captive chickens on the dough model. We first performed a G-test on the frequency of first attacks on different positions of the dough models by captive chickens against the null expectation. We then carried out a pairwise comparison between the frequency of first attacks on the head, mid-body and the tail.

## RESULTS

### Conspicuousness against leaf litter background

Modelling the conspicuousness of *U. liura* and *Teretrurus cf. sanguineus* according to avian visual systems indicate that the ventral colourations of both species are highly conspicuous against leaf litter background ($\Delta S > 3.00$) (Table 1). The results were consistent when accounting for the cone densities of VS (domestic chicken and peafowl) and UVS (blue tit) visual system of birds and against the average leaf litter of all transects (Table 1).

### Predation rates on snake models in field

All models were recovered after 86 h. However, many models ($N = 39$) were destroyed due to trampling by large mammals such as Gaur (*Bos gaurus*), Sambar deer (*Cervus unicolor*) and Asian elephants (*Elephas maximus*), and were, therefore, excluded from the analyses. A total of 50 models (i.e. 5% of all models) across both predation trials were found to have potential avian attacks. The GLMM analysis indicated that the number of attacks varied significantly among the treatments (Fig. 3) and that the model where predation rate was affected by phenotype (i.e. model colour) was significantly better than the null model where phenotype did not affect the predation rate ($\Delta$AIC = 16.4, $X^2$ = 24.402, $P < 0.0001$). The post hoc tests indicated that the number of attacks on the red (Estimate = −2.4349, $z$ = −3.371, $P < 0.01$) and yellow (Estimate = −1.7275, $z$ = −3.198, $P < 0.05$) models were significantly lower compared to that on the brown models. Attacks on the brown models did not differ significantly from that on either black (Estimate = 0.4834, $z$ = 1.328, $P$ = 0.65683) or novel coloured (Estimate = 0.6652, $z$ = 1.745, $P$ = 0.38739) models. Although the models with uropeltid colouration (red and yellow) experienced less predation compared to the novel models (orange), the difference was not statistically significant (red models: Estimate = −1.7697, 95% CI [−3.7821–0.2426],

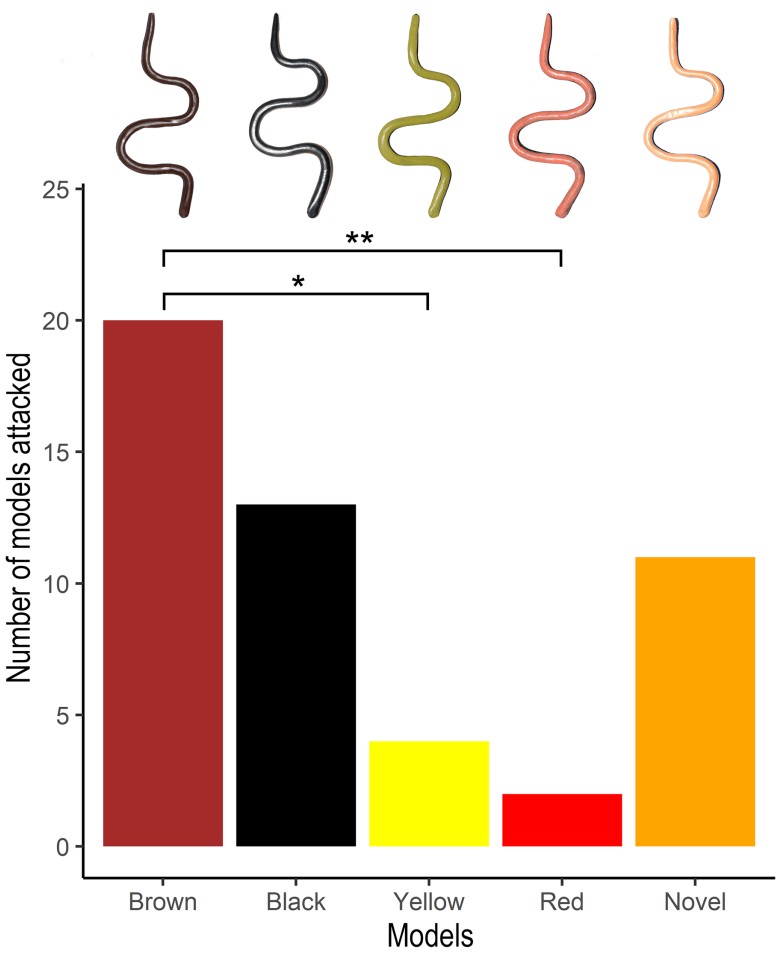

**Figure 3 Predation rates on snake models from field trials.** Number of attacks by avian predators on different snake models placed in the four transects in the BBTC tea plantations. Representative photographs of models of the five treatments (completely brown, completely black, black and yellow, black and red, and black and novel coloured) are shown above the respective bars. Numbers of asterisks indicate significance levels (*≤ 0.05, **< 0.01) between treatments based on Tukey post hoc tests.

$z = -2.369$, $P = 0.11423$; yellow models: Estimate $= -1.0623$, 95% CI [$-2.6060$–$0.4814$], $z = -1.854$, $P = 0.3248$).

### Captive bird experiments

A $G$-test on the attack position by the captive chickens on the dough models indicate that there was a significant difference in the position of first attacks ($G$-test: $G_2 = 17.102$, $P = 0.00019$). There were significantly higher number of attacks on the mid-body ($P = 0.00034$) and on the tail ($P = 0.0087$) compared to that on the head (Fig. 4).

### DISCUSSION

Our avian visual modelling indicates that the yellow and red ventral colourations of uropeltid snakes are highly conspicuous against the leaf litter background of all four transects and that uropeltid colours are readily visible to birds with both the VS and UVS

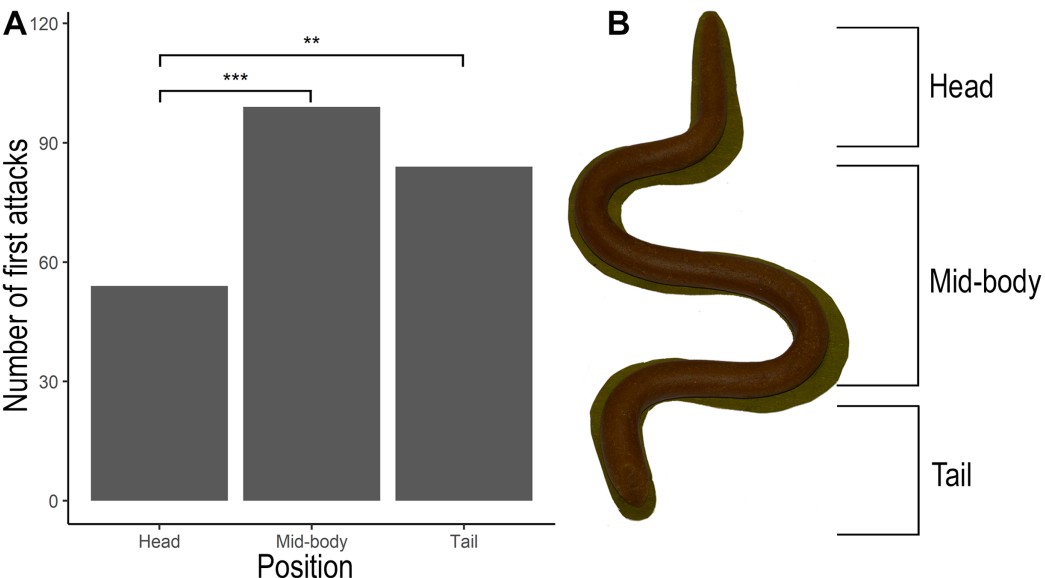

**Figure 4 Position of first attacks by captive chickens on edible dough models.** (A) Frequency of first attacks oriented towards different positions of the prey by captive chickens. (B) Dough model indicating the head, mid-body and tail region where the attacks were oriented. Numbers of asterisks indicate significance levels (**< 0.01, ***< 0.001) between treatments based on pairwise G-tests.

visual systems. The field experiments indicate that clay models with the yellow and red colouration of local uropeltid snakes experience reduced avian predation compared to the brown models. However, the novel coloured models (orange) did not differ from the brown or black ones. Thus, the results indicate that avoidance is not towards all conspicuous colours but specifically towards uropeltid colours, suggesting that colouration in uropeltid snakes may have evolved as an antipredatory defense mechanism against avian predators.

Colouration in uropeltid snakes has sometimes been attributed to coral snake mimicry (although not experimentally tested) (*Rajendran, 1985*). South Asian coral snakes (genus *Calliophis*) are venomous, semi fossorial elapids, which share similar habitats with uropeltid snakes. However, all known species of coral snakes found in India and Sri Lanka have red ventral surfaces while most species of uropeltids have varying patterns of yellow. Further, coral snakes in India are uncommon across their range (*Srinivasulu, Srinivasulu & Molur, 2014*). Several theoretical models and experimental studies have suggested that Batesian mimicry generally becomes more perfect when the model is rare (*Sherratt, 2002*; *Harper & Pfennig, 2007*; *Akcali & Pfennig, 2014*). Since coral snakes are uncommon, it is expected that selection would favour high precision of coral snake mimicry, which does not seem to be the case for uropeltids. Furthermore, some uropeltids with bright colouration are found outside the range of coral snakes, making Batesian mimicry unlikely (*Pfennig & Mullen, 2010*). Therefore, although coral snake mimicry cannot be completely ruled out in some species of uropeltid snakes with red ventral colourations, the Batesian mimicry hypothesis does not explain the evolution of yellow colouration present in most uropeltid snakes because would-be models are red.

Overall, our experiments suggest that the conspicuous ventral colouration in uropeltid snakes could have evolved as a warning signal towards avian predators. Given that these snakes are endemic to India and Sri Lanka, where they are protected under stringent laws, laboratory experiments with live animals were not feasible. Our experiment with captive chickens and dough models resembling uropeltid snakes show that attacks were more frequent on the tail than on the head (Fig. 4). This suggests that chickens can be deceived by the cephalic resemblance of the tail and direct their attacks towards the posterior regions, rather than towards the head as many predatory birds are known to do (*Smith, 1973*; *Curio, 1976*). Such a strategy that deflects attacks to the tail would also increase the handling time required to capture, kill and consume these snakes, making them unprofitable (*Humphreys & Ruxton, 2018*). Although we have not specifically tested whether deflection of attacks towards the tail increases handling time in uropeltid snakes, detailed observations on the sequence of events during predation of uropeltid snakes by junglefowls and peafowls indicate long handling time associated with deflected attacks towards the tail (*Gans, 1986*). *Gans (1986)* reported that ca. 95% of the attacks by fowls were oriented towards the rounded tail of uropeltid snakes. He noted that the birds took between 22 and 40 min to consume uropeltid snakes after multiple attacks to the tail. He also reports anecdotal evidence that junglefowls and spurfowls were more hesitant to attack the brightly coloured *Rhinophips drummondhayi*, than the cryptic unicoloured *R. philippinus*. These observations provide further support for the hypothesis that diverted attacks from the vulnerable head to the tail could potentially increase handling time needed to kill and consume uropeltid snakes, which birds could learn to associate with their ventral colourations.

Startle or deimatic displays involve behaviours wherein conspicuous colours are suddenly displayed, providing prey a survival advantage (*Umbers, Lehtonen & Mappes, 2015*; *Umbers & Mappes, 2015*). Although there has been disagreement regarding what constitutes a deimatic display and how it differs from aposematism (*Ruxton, Sherratt & Speed, 2004*; *Umbers, Lehtonen & Mappes, 2015*; *Umbers & Mappes, 2015*, *2016*; *Skelhorn, Holmes & Rowe, 2016*), there is general agreement that deimatism involves a momentary transient display of conspicuous signal (*Ruxton, Sherratt & Speed, 2004*; *Olofsson et al., 2012*; *Umbers & Mappes, 2016*) that triggers an unlearnt avoidance response in predators (*Umbers et al., 2017*, *2019*). While uropeltid colours could also have a deimatic function, our experiments show that birds avoided uropeltid snake models that were static (pinned to the ground) and did not show any momentary display of conspicuous colourations. The snake models in our field experiment were painted such that the conspicuous colours were visible on the lateral side and thus were displayed and visible to small predators throughout the experimental duration (Fig. S2). Further, the preferential avoidance of only uropeltid colours and not all conspicuous colours suggests that this avoidance is learnt. However, as we did not design our experiments to test for a deimatic function of uropeltid colourations, we cannot preclude the possibility that uropeltid colours could also function as a startle display.

Advertising increased handling time due to deflection of attacks would, however, be advantageous only when alternative prey are abundant and when predators do not learn to

recognise the deception. Repeated exposure to deflective structures and behaviours would reduce the effectiveness of deceiving predators. However, although birds regularly feed on venomous and non-venomous snakes (*Guthrie, 1932*), there is a risk associated with attacking snakes since many snakes are venomous. Hence, birds tend to orient their attacks towards the portion of the body with a bilateral indentation (neck) or towards the region with eye-like markings (*Smith, 1973*; *Curio, 1976*). The costs associated with attacking venomous snakes may prevent birds from learning the cephalic resemblance of the tail in uropeltid snakes. Further, it has been shown that the speed of a predator's learning varies for different traits (*Chittka & Osorio, 2007*; *Balogh et al., 2010*), especially when other prey are available in the community (*Kikuchi et al., 2019*). For instance, animals show higher rates of learning towards colour signals than towards patterns or shapes (*Bain et al., 2007*; *Aronsson & Gamberale-Stille, 2008*, *2012a*; *Kazemi et al., 2014*; *Sherratt et al., 2015*). Given that colours are more salient than other cues, birds may be able to associate colour with increased handling time in uropeltid snakes faster than learning the cephalic resemblance of the tail. Also, deflective traits are thought to be associated with life histories that reduce exposure to predators, thereby reducing the potential of learning such deceptive traits (*Humphreys & Ruxton, 2018*). Uropeltid snakes are highly seasonal and are active above ground or near the surface during the monsoons (*Rajendran, 1985*) when the species richness and abundance of leaf litter arthropods are high (*Janzen & Schoener, 1968*; *Frith & Frith, 1990*; *Develey & Peres, 2000*). High densities of arthropods during the wet seasons in tropical regions could serve as an alternative prey base for avian predators. The short activity period of uropeltid snakes would also reduce the ability of avian predators to learn to ignore deflective traits.

Although we cannot completely rule out the possibility of Batesian mimicry in some species of uropeltids, our experiments together support the hypothesis that the conspicuous colourations in these snakes act as warning signals against avian predators. Our results are also consistent with the novel hypothesis that conspicuous colourations in uropeltid snakes have an antipredatory function based on advertising long handling time associated with diverted attacks to the tail. Nonetheless, given that we have not explicitly tested whether misdirected attacks to the tail increases the handling time of these snakes, we acknowledge that a functional relationship between the colouration and tail shape and its role in reducing predation cannot be ruled out. For instance, many species of uropeltid snakes possess yellow or red blotches or stripes on the lateral sides of the tail (Fig. 2B; Figs. S1D and S1F) that could divert the attention of predators and deflect attacks towards the tail. However, further experiments would be required to determine whether colouration and tail shape interact to divert attacks and how this would influence handling time and learning in predators.

Antipredatory defenses involving both deflection of attacks and warning colourations may not be restricted to Uropeltidae. Fossorial reptiles, especially snakes, tend to have long trunks and short rounded tails (*Wiens & Slingluff, 2001*; *Wiens, Brandley & Reeder, 2006*), the latter of which many species display to divert attacks from the head to the tail (*Greene, 1973*, *1979*, *1988*; *Han & Young, 2018*). Several fossorial snakes (e.g. Cylindrophidae, Anomochilidae, Anillidae, Atractaspinae, Aparallactinae) exhibit

conspicuous colourations potentially having a similar function as in uropeltid snakes. However, how widespread such strategies are across animals needs to be evaluated.

## CONCLUSION

Our study highlights how combinations of defensive strategies such as deflection, which increases handling time, and warning signals, could function synergistically against predation. Different antipredatory strategies need not be mutually exclusive and can interact depending on the predators' visual perception (*Stevens, 2007*). For instance, it has been shown that patterns with strong internal contrast can increase conspicuousness (*Aronsson & Gamberale-Stille, 2012b*) but can also generate a disruptive effect (*Stevens & Cuthill, 2006*; *Schaefer & Stobbe, 2006*). Warning colourations can also be distance dependent, providing camouflage at greater distances while being conspicuous at close proximity (*Tullberg, Merilaita & Wiklund, 2005*; *Barnett, Cuthill & Scott-Samuel, 2018*). For instance, the contrasting bands of coral snake mimics and the zig-zag pattern of many vipers act as warning signals to predators (*Pfennig, Harcombe & Pfennig, 2001*; *Wüster et al., 2004*; *Niskanen & Mappes, 2005*) but these bands can also function in camouflage during motion through the flicker-fusion effect (*Lindell & Forsman, 1996*; *Titcomb, Kikuchi & Pfennig, 2014*). Despite considerable advances in our understanding of animal defenses, we are yet to completely understand how different strategies interact with each other and further experiments are necessary to understand under what conditions such interactions are advantageous against predation.

## ACKNOWLEDGEMENTS

We thank Umesh P.K. and BBTC tea estates, Manjolai for allowing us to carry out our field experiments in their tea plantations. We thank John Endler for his comments on an early draft of the manuscript. We thank David Gower for his discussions on uropeltid snakes. We thank Almut Kelber and Balamurali G.S. for the discussions on avian visual models; Udita Bansal, Harshad Mayekar, Sairandhri Lapalikar, Ashish Nerlekar, Rishiddh Javery, Alex Johny, Sreejit Allipra, Gopal Murali, Gayatri Kartha, Amal Reji, Divya P.S., Midhun Krishna, Meenakshi Jyothis, Maria Babu, Varun Kher, Jewel Johnson and Reshnu Raj for assisting with preparations of the snake models; Umesh P.K., John, Alex Johny, Udita Bansal, Sreejit Allipra, Varun Kher and Jayasooryan C.S. for assisting during fieldwork in BBTC. We also thank Nawaf Abdul Majeed and Rohit Anand for their help with preparing the dough models and conducting the captive chicken experiments. We thank Umesh P.K. for images of the ventral colouration of uropeltid snakes.

### Funding

This work was supported by an INSPIRE Faculty Award to UK (DST/INSPIRE/04/2013/000476) and intra-mural funding from IISER Thiruvananthapuram. The funders had no role in study design, data collection and analysis, decision to publish, or preparation of the manuscript.

## Grant Disclosures

The following grant information was disclosed by the authors:
INSPIRE Faculty Award to UK: DST/INSPIRE/04/2013/000476.
Intra-mural funding from IISER Thiruvananthapuram.

## Competing Interests

The authors declare that they have no competing interests.

## Author Contributions

- Vivek Philip Cyriac conceived and designed the experiments, performed the experiments, analysed the data, prepared figures and/or tables, authored or reviewed drafts of the paper, approved the final draft.
- Ullasa Kodandaramaiah contributed reagents/materials/analysis tools, authored or reviewed drafts of the paper, approved the final draft.

## Animal Ethics

The following information was supplied relating to ethical approvals (i.e. approving body and any reference numbers):

All applicable international, national and/or institutional guidelines for the care and use of animals were followed. The experimental protocol was approved by the Institutional Animal Ethics Committee of Indian Institute of Science Education and Research Thiruvananthapuram.

## Field Study Permissions

The following information was supplied relating to field study approvals (i.e. approving body and any reference numbers):

The field experiments were done with the permission of Bombay Burma Trading Corporation, Limited, which owns the lands.

## Data Availability

The raw data is available at Figshare: Cyriac, Vivek; Kodandaramaiah, Ullasa (2019): Conspicuous colours reduce predation rates in fossorial uropeltid snakes. figshare. Dataset. https://doi.org/10.6084/m9.figshare.7957223.v1.

## Supplemental Information

Supplemental information for this article can be found online at http://dx.doi.org/10.7717/peerj.7508#supplemental-information.

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
