# Peer review of "Conspicuous colours reduce predation rates in fossorial uropeltid snakes"

_PeerJ, doi:10.7717/peerj.7508_

## Round 0.1 · original submission · Major Revisions

Dear Drs. Cyriac and Kodandaramaiah:

Thanks for submitting your manuscript to PeerJ. I have now received two independent reviews of your work, and as you will see, the reviewers raised some concerns about the research. Despite this, these reviewers are optimistic about your work and the potential impact it will have on research pertaining to predation avoidance mechanisms in uropeltid snakes. Thus, I encourage you to revise your manuscript accordingly, taking into account all of the concerns raised by both reviewers.

Please address the concerns for a lack of sufficient literature. Please also discuss in your revision a deimatic signal function of the tail display (per reviewer 1). Please also consider alternative hypotheses and how to best rule them out (per reviewer 2).

I look forward to seeing your revision, and thanks again for submitting your work to PeerJ.

Good luck with your revision,

-joe

·

Basic reporting

The manuscript is overall well written and addresses an interesting question concerning an unstudied, conspicous color signal found in fossorial snakes. The authors, however, make insufficient use of references and have missed out on important literature from recent years relevant to their publication (examples and suggestions provided in general comments).
What is missing to make this a great manuscript is any information/data/observation on how much longer the handling time really is due to the hardened tail of these snakes. Further, figures should be of the same format and the authors should get rid of some redundancies within the manuscript (see general comments).

Experimental design

The design of the study is sufficient in respect to the hypothesis, however, data on avoidance learning caused by longer handling times (due to the snake’s hard skin shield) as well as data on handling times would significantly improve the overall work.

Validity of the findings

Findings of the study are valid and contribute to a better understanding of unexplained conspicuous coloration in animals.

Additional comments

I do like this manuscript and enjoyed reading about the authors' work. I commend the authors on their study. The study system is very interesting and the authors deliver a good approach to answer their hypothesis. Nonetheless, I have made a number of comments that the authors should consider:
L 37-38 snakes are only (mostly) brightly colored ventrally, correct? If so, please add “ventrally”
L 38-39 the sentence about Batesian mimicry could be discarded here as it might not be necessary in the abstract
L 65 & 67 I think there could be more general (classical) literature added (e.g. Poulton 1870, Cott 1940, Edmunds 1974, etc..)
L 68 give examples
L 69 examples and potentially literature on unexplained or unstudied conspicuous signals. This is extremely interesting and more research is being done on new systems. (for example, and see literature therein: Rößler et al. 2019)
L 70 suggestion: conditions devoid of light
L 71-72 mention examples of snakes (other than your study species)
L 73 “are visual stimuli”… to whom? Only to animals who can perceive this. Make this clearer here.
L 73 …“costly to produce”… This depends strongly on the color. There is a great difference between structural colors and pigments, and between different pigments.
L 79-81 please add information whether these species are diurnal or nocturnal (if information is present). And are they more likely to come to the surface at night or during the day?
L 84 is there any literature on tail morphology of this genus. If so, please cite.
L 84-86 what you describe here is a perfect description of a startle display or deimatic display. I wonder why you have not mentioned this term once in your manuscript. At least there should be a discussion whether this behavior coupled with the coloration could be deimatic (see Umbers et al. 2017, 2015, Umbers & Mappes 2015)
L 88-89 this seems a bit speculative. I imagine a hungry bird still being able to eat the snake, similarly other animals. Why would they keep directing their attack towards the tail?
L 98 main predators: is there information (observation/data/literature) on this? Or do you only assume birds are main predators? What about rodents? In case there is no info, please be more cautious with the phrasing
L 103 again, please use more cautious phrasing, you conclude that they have “evolved as a defense”, but your experiments, even though significant, are just models, you have no information on real predation (see Bateman et al. 2017, Rößler et al. 2018). You also have no information if coloration first evolved with different purposes (thermoregulation, immunobiology..) and later, secondarily, acquired an anti-predator function
L 108 insert space after 34
L 124 how were “representative leaves” chosen? This sounds very subjective
L 128-129 why did you not take measurements from the dorsal surface of the snakes. In the picture of figure 2 (the tail display), clearly the internal contrast, the conspicuous color builds against the dorsal color seems a lot more informative. I don’t disagree about the conspicuousness, but visual models, as nice as they are, really just compare two colors with each other, but detection and conspicuousness in complex environments are more multifaceted than a contrast. Just be cautious about this and the conclusions you draw from visual models
L 141 “contrasts” instead of “colours”
L 145 you mean Weber fraction?
L148 delete bracket
L 132-151 the visual modeling seems very thorough. Well done.
L 154 please clarify: a total of 500 models? Total of 1000 models? (both predation trials?)
L 161 you mention details for the acrylic paint here, why not for the clay product? I know it is in your supplements, but I think it is important to mention the clay product and brand in the main manuscript (again, see Bateman et al. 2017 & Rößler et al. 2018).
L 168 Cite literature and explain how you identified avian attack marks. (e.g. V-shaped or U-shaped marks, Willink et al. 2014)
L 169-170 why two different seasons? Did you check your data for effects? Also predators might be seasonally active, possibly skewing your data..
L 175 “with dough models edible to chickens resembling …” is more or less repeated in the next sentence again. Discard one.
L 178 the yellow and brown paper should be explained. When was which used and why?
L 185 delete bracket
L 196/197 G test & G-test. Please unify throughout the manuscript
L 204-205 please specify in which analyses post-hoc tests for multiple comparisons were conducted. It is not clear from this paragraph
L 219 significantly
L 241 redundancy of “position of first attacks”
L 242 significantly
L 255 may have evolved
L 264 double would
L 278-280 I am not convinced that the handling time would be increased to the extent of stopping an attack. I think deimatic display should be discussed here, including further examples of tail displays in other animals (e.g. Taricha granulosa).
L 280 please add literature on deflection strategies
L 291 I think there is a word missing after “however”.. can be?
L 302-305 split in two sentences. Again, a word missing, after “support”
L 313-328 very nice paragraph, good use of literature!
Literature: check your literature for genus and species names and use italics (Dendrobates pumilio, Parus major etc…). also check Gans 1986, here you use the journal abbreviation

Figure 4A: why use a different bar chart format here? Be consistent. So, either use the same as in Figure 3 or change Figure 3 to the format of Figure 4A. (use same label of y-axis, grid vs. no grid, p-values vs. asterisk..)

Figure 4B: the snippet of the snake looks a bit sloppy. If possible, either remove the dark background thoroughly, use a different photo, or draw up a simple illustration.
The following bit might be a bit pedantic (feel free to ignore): Use a different illustration of body regions than the curly brackets you used (Even square brackets are okay, either way, please make sure to align them)

You do not mention the supplementary tables S1 and S2 within the text, and I am not entirely sure what they contain (especially S1), as the caption is not really explaining it well. Please explain these supplements and refer to them in the text if needed.

General: check your manuscript for double spaces (“ “), I think I saw some.

References I mentioned in my comments:
Bateman, P.W., Fleming, P.A., Wolfe, A.K., 2017. A different kind of ecological modelling: the use of clay model organisms to explore predator--prey interactions in vertebrates. Journal of Zoology, 301(4), 251–262. https://doi.org/10.1111/jzo.12415

Cott, H.B., 1940. Adaptive coloration in animals, Vol. 540. Oxford, England: Oxford Adaptive coloration in animals.

Edmunds, M., 1974. Defence in animals: a survey of anti-predator defences. Burnt Mill: Longman.

Poulton, E.B., 1870. The colours of animals: their meaning and use, especially considered in the case of insects, p. 400. New York: D. Appleton.

Rößler. D.C., Pröhl, H, Lötters, S. 2018. The future of clay model studies. BMC Zoology 3:6. https://doi.org/10.1186/s40850-018-0033-6

Rößler, D.C., Lötters, S., Mappes, J., Valkonen, J., Menin, M., Lima, A.P., Pröhl, H. 2019. Sole coloration as an unusual aposematic signal in a Neotropical toad. Scientific Reports. https://doi.org/10.1038/s41598-018-37705-1

Umbers, K.D.L., De Bona, S., White, T.E., Lehtonen, J., Mappes, J., Endler, J.A., 2017. Deimatism: a neglected component of antipredator defence. Biology Letters, 13(4). https://doi.org/10.1098/rsbl.2016.0936

Umbers, K.D.L., Lehtonen, J., Mappes, J., 2015. Deimatic displays. Current Biology: CB, 25(2), R58–R59. https://doi.org/10.1016/j.cub.2014.11.011

Umbers, K.D.L., Mappes, J., 2015. Postattack deimatic display in the mountain katydid, Acripeza reticulata. Animal Behaviour, 100, 68–73. https://doi.org/10.1016/j.anbehav.2014.11.009

Willink, B., García-Rodríguez, A., Bolaños, F., Pröhl, H., 2014. The interplay between multiple predators and prey colour divergence. Biological Journal of the Linnean Society. Linnean Society of London, 113(2), 580–589. https://doi.org/10.1111/bij.12355

Reviewer 2 ·

Basic reporting

No comment

Experimental design

No comment

Validity of the findings

No comment

Additional comments

Review of Uropeltid snake colors

This is a lovely study with excellent methods and scholarship, but I’m not convinced that the authors have ruled out simpler alternative hypotheses to the one that they present in this manuscript. It needs either to be rewritten, or more data. My detailed comments are below.

63-68: This is actually too broad – could be cut
72-73: Colors aren’t necessarily costly to produce if their structures have some other function. After all, an organism has to have some sort of spectral reflectance distribution
95-104: This hypothesis has strange logic. The cephalized tail is hypothesized to deceive predators, yet the warning coloration is hypothesized to alert predators to this deception.
145: Weber
171-186: It seems that the most appropriate experiment to test whether a cephalic tail increases handling times would be pastry snakes with a distinct head and tail, and pastry snakes with two heads. Birds can certainly learn different handling times in general – there is a mountain of literature to show this. Furthermore, many studies with artificial snakes in the field have shown that attacks are directed towards the heads of snakes rather than the tails. It’s not easy to see what is gained from this experiment.
195: Only a G-test or the GLMM should be used, to present both is redundant. It should really be the GLMM since it reflects the design of the experiment.
229: P-value comes from the Likelihood Ratio Test? Please specify
234 – 238: Isn’t this saying the same thing? Confusing
265: Furthermore
267: Pfennig & Mullen PRSB would be relevant to cite here
284 -290: would be very relevant to have in the Introduction. Also, what were the handling times for snakes that were attacked at the head?
291-301: The major problem that I have with this hypothesis is that to learn that uropeltid coloration signals long handling times, birds would have to encounter multiple snakes, yet encountering multiple snakes makes them more likely to learn to distinguish head from tail. The distinctive colors of uropeltids might then even act as a cue to alert predators to the possibility of the false head. In fact, it is hard to imagine that this is something that the birds wouldn’t learn. To be convinced of the plausibility of this paper’s hypothesis, I would need to see that cephalized tails increase handling times, and that avoidance of colors associated with long handling times is learned more quickly than distinguishing false heads from real heads. Additionally, it would be interesting to see if there is an interaction between tail shape and coloration on attack probability. At present, it is more parsimonious (and equally consistent with the data) to conclude that birds have an innate aversion to uropeltid colors, and that the cephalization of uropeltid’s tails misdirects predator attacks, but that there is not an interaction between the two strategies.

---

## Round 0.2 · Minor Revisions

Dear Drs. Cyriac and Kodandaramaiah:

Thanks for re-submitting your manuscript to PeerJ. I have now received two independent reviews of your work (both from the previous reviewers), and as you will see, the reviewers still have some concerns about the research. Note that both reviewers still feel that the English and grammar can be improved throughout the manuscript, so please do so in your revision.

Please consider your hypothesis testing based on your experimentation (per reviewer 2’s suggestions). Also ensure that your methods are clearly described, and that justification is provided for all methodological approaches and comparisons (statistics).

Accordingly, please address the issues raised by both reviewers and submit another revision. I am sure this third revision will bring you close to acceptance for publication.

I look forward to seeing your revision, and thanks again for submitting your work to PeerJ.

Good luck with your revision,

-joe

·

Basic reporting

I’m happy to see that the authors have significantly increased the use of literature as well as applied the reviewers’ suggestions on several aspects of the discussion, improving the overall manuscript.

Yet, I have a few minor comments (Lines refer to the document including track changes). You can find these under the general comments.

Experimental design

see general comments

Validity of the findings

see general comments

Additional comments

L. 90 mornings
L. 323-338 I appreciate your inclusion of discussing a deimatic signal function, however, I think that you are too concerned in this paragraph about comparing this function to what you did in your experiment. I do not think this is necessary. Your models were neither moving, nor in the potential startle position, thus, there is no need to compare it, because you simply did not test any deimatic function with your models. I just think, based on the photograph, it is important to mention and discuss the potential presence of deimatism, but be clear that this was not what you tested (or intended to test). I disagree that the avoidance of your static models by birds conveys any information about a potential startle function.
L. 341 be consistent about your use of British English, thus: behaviour
L. 342-344 This sentence is a bit clumsy, can you rephrase?
L. 344 Hence,
L. 344 has
L. 350 shapes
L. 352 be careful about the use of “cue” vs. “signal”. You use “colour cue” here. Be sure to know the difference and use the one you really mean.
L. 355 repetition of predator

Reviewer 2 ·

Basic reporting

see below

Experimental design

see below

Validity of the findings

see below

Additional comments

Review of Uropeltid Snake Colors Revision

The authors have made revisions that greatly improve the manuscript. Still, I am concerned that the hypothesis they advocate for is not the only hypothesis that should be considered to explain the coloration of these fossorial snakes, because given the data available it is not possible to reject simpler alternatives. I think that the presentation needs to be changed to better reflect the information we have gained from the experiments that were done, i.e. which hypotheses were rejected by the data consideration.

1) I do actually insist that the authors get rid of the G-squared test where the GLMM can be substituted for it; presenting methods used in other studies only makes sense if parameter estimates and their standard errors are going to be extracted and compared. The G-squared test doesn’t provide this information, and is more likely to produce spurious results because it doesn’t account for the structure of the data.
2) The authors make persuasive arguments in the new Discussion for the plausibility of their hypothesis, but even those these new arguments are very interesting, they don’t add new data that reject other hypotheses. So this doesn’t really address the comments that I made on the earlier draft. It does seem that the data can reject Batesian mimicry, since both yellow and red were protected when local coral snakes are red, but how can one reject the hypotheses that a) tail shape and color operate independently, or that b) tail shape and color interact to misdirect attacks towards the tail (when the tail is flipped upside down, as in Figure 2), but that they aren’t actually signaling longer handling times? The replicas used in the field don’t seem to display the natural defensive positions of these snakes, if Figure 2 is representative, making it even more difficult to conclude exactly what the functional relationship between tail shape and coloration is. I want to be clear that I think the authors have every right to propose the hypothesis that they propose, but I don’t think it is consistent with the scientific method to present it as the hypothesis that is best supported by the data. We trust the hypotheses that we have attempted to reject the most rigorously, and fail to reject after other contenders have been eliminated.
3) I didn’t catch this on the first read-through, but in the field experiments, are attacks more frequently directed towards the heads, tails, or midsections? Is the pattern consistent with the captive chicken experiment?

261-265: This is a bit odd – if there is no significant difference, it is more accurate to say that the groups were not different.

---

## Round 0.3 · accepted · Accept

Dear Drs. Cyriac and Kodandaramaiah:

Thanks for revising your manuscript to PeerJ, and for addressing the concerns raised by the reviewers. I now believe that your manuscript is suitable for publication. Congratulations! I look forward to seeing this work in print, and I anticipate it being an important resource for research on predation avoidance mechanisms in uropeltid snakes.

Thanks again for choosing PeerJ to publish such important work.

-joe

# ·

Basic reporting

I think the authors sufficiently addressed the concerns raised by the reviewers and revised their work where possible. While there might remain shortcomings in experimental design (which I know are not revisable at this point), I think this study is ready to be published.

Experimental design

see above

Validity of the findings

see above

Additional comments

see above

Reviewer 2 ·

Basic reporting

See Below

Experimental design

See below

Validity of the findings

See Below

Additional comments

The authors have addressed my concerns. I don't see any reason to hold this paper up.